# Fertilizer Application in Contract Farming: A Risk Analysis

**Yiming Zhang [1], Rui Yang [1], Kai Zhao [2,\*] and Xiangzhi Kong [1,\*]**

[1] School of Agricultural Economics and Rural Development, Renmin University of China, Beijing 100872, China; zhym@ruc.edu.cn (Y.Z.); 2018102686@ruc.edu.cn (R.Y.)

[2] Institute of Animal Science, Chinese Academy of Agricultural Sciences, Beijing 100193, China

\* Correspondence: 2019000658@ruc.edu.cn (K.Z.); kongxz@ruc.edu.cn (X.K.); Tel.: +86-18811798766 (X.K.)

**Abstract:** To our knowledge, this is the first study in the world to incorporate risk into the contract-farming participation decision process and to examine empirical evidence on the effect of contract farming on fertilizer application, filling the research gaps in the relevant literature and providing perspectives on both chemical fertilizer reduction and organic fertilizer application behavior. To estimate the impact of contract farming on farmers' application of chemical fertilizer and organic fertilizer, we used data on tea farmers from the Fujian and Hubei provinces in China and the recursive binary probit model. The 2SLS (two-stage least-squares) model was used to estimate the impact of the contract-farming participation rate on farmers' organic and chemical fertilizer application intensity. The empirical results show that farmers' risk-prevention ability had a significant negative impact on farmers' contract-farming participation decision and rates. Contract-farming participation had a significant, positive impact on farmers' organic and chemical fertilizer application intensity, while contract-farming participation rates had a significant, negative impact on the intensity of chemical fertilizer application by farmers. However, contract-farming participation rates did not have a significant impact on organic and chemical fertilizer application intensity. To promote fertilizer reduction and organic fertilizer incremental application, an effective strategy could be to promote farmers' participation in contract agriculture.

**Keywords:** contract farming; organic fertilizer; chemical fertilizer; risk





## 1. Introduction

In recent years, contract farming has come into sharp focus, particularly in developing countries [1–3]. It offers a crucial mechanism for integrating smallholder farmers into modern agriculture—a feature that is more prominent in developing countries than in developed nations [4–7]. There are several benefits of contract farming, including economic development, a rise in farmers' welfare, a reduction in food insecurity, and an increase in agricultural productivity [8–11]. Furthermore, it offers superb opportunities for the stewardship of the environment and sustainable farming practices [12]. These aspects significantly contribute to achieving multiple global sustainable development goals (SDGs), primarily those related to no poverty, zero hunger, clean water and sanitation, and good health and well-being [13]. One such opportunity lies in the potential of contract farming to influence the application of chemical fertilizers—an area of increasing importance given the environmental and health implications associated with excessive chemical fertilizer use and reduced chemical fertilizer use [14]. According to Food and Agriculture Organization (FAO) statistics, in 2020, the intensity of chemical fertilizer application measured according to potassium content in China was 61.92 kg/ha, which was much higher than the value of 20.23 kg/ha in the European Union (EU), and the intensity of organic fertilizer application measured according to potassium content was 29.72 kg/ha, which was lower than the value of 39.59 kg/ha in the EU (data source: Food and Agriculture Organization (FAO); https://www.fao.org/faostat/en/#data/ESB (accessed on 19 July 2023)). In light of contract farming's benefits and potential to influence sustainable farming practices, we can focus on its role in reducing chemical fertilizer use.

The worldwide emphasis on sustainable agriculture is driven by the growing understanding of the environmental, health, and economic implications of traditional farming methods, among which is, notably, the excessive use of chemical fertilizers [15]. These chemical fertilizers, while boosting crop yield in the short term, have been linked to a variety of environmental issues, such as soil degradation, water pollution, and the loss of biodiversity. Moreover, overreliance on chemical fertilizer often results in diminishing returns over time, with increased quantities being needed to maintain productivity [16]. Consequently, the adoption of organic fertilizer and sustainable farming techniques is being advanced as an essential step toward sustainable agriculture [17]. In recent years, the potential of contract farming to foster sustainable agricultural practices has been increasingly recognized [18]. Given the pivotal role that contract farming plays in shaping farming practices, it is worth exploring how this model can influence the transition towards reduced chemical fertilizer usage and increased organic fertilizer application.

The limited available literature explores the effects of contract farming on chemical and organic fertilizer application, and the role of risk in this context. Ragasa et al. (2018) conducted empirical research demonstrating how contract farming can increase the adoption of new technologies and improve crop yields by providing an assessment of various maize-based contract-farming schemes in Ghana [18]. Schewe and Stuart (2017) presented a mixed-methods study on seed corn contract farming in southwest Michigan, revealing that competitive agricultural contracts pose significant structural barriers to adopting climate change mitigation behaviors, including constraints on information access and decision making [12]. Ren et al. (2021) conducted an empirical study on 623 Chinese farmers, finding that contract farming increases the likelihood of applying environmentally sustainable control technologies and organic fertilizers, thereby promoting sustainable agricultural practices and potentially mitigating the overuse of organic fertilizers [19]. Gao et al. (2022) developed a dynamic analysis framework to assess the impact of contract farming on farmers' use of organic fertilizers, finding empirical evidence from a survey of 473 vegetable farmers in Shandong, China, indicating that contract-farming participation increases the probability of organic fertilizer application by 50.7% [20]. Mishra et al. (2018) provided empirical evidence showing that contract farming in baby corn production leads to higher yields and the reduced use and cost of chemical fertilizers without compromising yield, effectively enhancing the livelihood of smallholders and reducing environmental degradation [21]. The reviewed literature offers valuable insights for the current study, although certain limitations should be noted.

The shortcomings of the existing literature are mainly due to the following three factors: Firstly, while existing research acknowledges risk as a determinant in farmers' adoption of technology, many studies merely reduce this risk to a matter of preference and choice, a subjective measure lacking standardized quantification. Secondly, a common limitation of existing studies is their emphasis on farmers' willingness to adopt technology as the dependent variable while seemingly disregarding the potential disparity between expressed willingness and actual behavior. Lastly, the prevailing focus in the existing literature either explores the influence of risk on farmers' participation in contract farming or assesses the impact of contract farming on organic and chemical fertilizer application behavior. These studies, however, do not incorporate risk, contract farming, and chemical fertilizer application behavior into a unified analytical framework. Consequently, they do not elucidate the potential mediating role of risk in the impact of contract farming on farmers' organic fertilizer application and chemical fertilizer usage reduction practices.

This paper attempts to compensate for the shortcomings of the existing literature by exploring farmers' fertilizer and organic fertilizer application behavior. This study explores the impact of contract farming on the reduction in the usage of chemical fertilizer and the increased application of organic fertilizer in the case of China, and how farmers' risk-prevention ability impacts the contract-farming participation decision of farmers. Our research aims to shed light on how contract farming can catalyze more sustainable and environmentally friendly agricultural practices, leading to a healthier ecosystem and better

crop yield in the long term. This study utilized micro-level survey data on tea farming households in the main tea-producing provinces of Fujian and Hubei, collected by our research team in 2016. Initially, this study focused on gauging risk through the lens of farmer's risk-prevention ability, as opposed to their risk preferences. Subsequently, it incorporated the actual patterns of organic and chemical fertilizer usage by farmers as the dependent variable. Utilizing a binary recursive probit model, this study estimated the impact of contract-farming participation decision and rate from a risk perspective on the decision to use organic and chemical fertilizers and the intensity of organic and chemical fertilizer usage by farmers. The 2SLS (two-stage least-squares) model was employed to check the robustness of the results. By integrating risk into the decision-making process of contract farming and investigating its impact on farmers' chemical fertilizer application behavior, this study aims to contribute to the advancement of sustainable agriculture practices and provide empirical evidence to support policy making related to the promotion of contract farming.

## 2. Theoretical Framework

This section presents a theoretical framework to understand the relevance between contract-farming participation and the use of chemical and organic fertilizers. Considering the time-dependent impact of organic fertilizer on soil quality, this study analyzes how contract farming influences farmers' organic fertilizer use in a dynamic context, with soil quality equilibrium as a transversality condition. Our study treats contract-farming participation as endogenous, with heterogeneous farmer characteristics. It also incorporates farmers' risk-prevention ability, which is often overlooked in the literature, as a key factor in their decision to participate in contract farming. Assuming that farmers decide between organic and chemical fertilizers for each land unit, with these inputs being substituted, the study posits risk mitigation as a crucial farm-level characteristic. Enlightened by the practice of Ma et al. (2018) [22] and the location models proposed by Fulton and Giannakas (2013) [23], the cost function is expressed as follows:

$$\begin{aligned}
C = C^O + C^M &= (\alpha - \beta\theta^O)O(t) + (\alpha - \beta\theta^M)M(t) \\
&= (\alpha - \beta\eta\widetilde{e}^O(z))O(t) + (\alpha - \beta\eta\widetilde{e}^M(z))M(t)
\end{aligned} \tag{1}$$

where $\alpha, \beta > 0; \alpha - \beta\theta > 0; \theta^O > \theta^M$. (Throughout the text, the superscripts $O$ and $M$ refer to the application of organic and chemical fertilizers, respectively; the superscripts $C$ and $N$ refer to participation and non-participation in contract farming, respectively.) Let $C^O$ and $C^M$ denote the cost of chemical fertilizer application and organic fertilizer application, respectively. Let $\theta = \eta\widetilde{e}(z)$ denote farmer characteristics, where $\eta$ denotes household-level and farm-level characteristics. Let $\widetilde{e}(z)$ denote the risk faced by a farmer, which decreases with the farmer's risk-prevention ability. As opposed to chemical fertilizers, suppose that organic fertilizers pose a higher risk in the short term, but they improve soil quality in the long run, which implies $\widetilde{e}^O > \widetilde{e}^M$, so $\theta^O > \theta^M$. Let $\theta$ be distributed over the interval $[0, 1]$, with $\theta = 0$ implying the lowest impact and $\theta = 1$ implying the highest impact on net returns from production. $O(t)$ and $M(t)$ indicate a farmer's organic and chemical fertilizer expenditures, respectively, at time $t$. Let us assume that the farmer characteristics ($\theta$) decrease with the cost function, which implies $C_\theta(\cdot; \theta) < 0$ and $C_{\theta\theta}(\cdot; \theta) > 0$. (In this paper, the subscript represents the partial derivative). The cost functions are linear in $O(t)$ and $M(t)$, so the marginal costs of chemical fertilizer and organic fertilizer, $C_O$ and $C_M$, respectively, only depend on $\theta$. (To simplify the expressions, $t$ is omitted unless necessary.)

Let us presume that farmers do not change their decision to participate in contract farming. The income function is given as:

$$R = R^O + R^M = PY^O(M, O, S; \theta^O) + PY^M(M, O, S; \theta^M) \tag{2}$$

Let $Y(\cdot)$ denote the production function of one unit of land cultivation and $Y(\cdot)$ be strictly concave in arguments $O, M, S$ and additively separable. Let $Y^O$ and $Y^M$ denote the production function of one unit of land cultivation when organic fertilizer and chemical fer-

tilizer, respectively, are applied. Let $R^M$ and $R^O$ represent income from sales of agricultural products when chemical and organic fertilizers, respectively, are applied. (On the premise of not losing generality, we only consider the impact of risk perception on output and cost and do not consider the changes in product price that may be caused by changes in product quality. Nevertheless, this does not change the results of the analysis.)

The profit function of farmers applying chemical fertilizer can be expressed as:

$$profitM = R^M - C^M = PY^M(M, S; \theta^M) - (\alpha - \beta\theta^M)M(t) \tag{3}$$

The profit function of farmers applying organic fertilizer can be expressed as:

$$profitO = R^O - C^O = PY^O(O, S; \theta^O) - (\alpha - \beta\theta^O)O(t) \tag{4}$$

Let $C^M$ and $C^O$ represent the costs of chemical and organic fertilizer application per unit area of land cultivated, respectively. Figure 1 shows the cost and benefit functions of non-contract agricultural farmers as functions of $\theta$. Chemical fertilizers are not profitable when $\theta < \underline{\theta}^M$, but they are profitable when $\theta > \underline{\theta}^M$. Organic fertilizers are not profitable when $\theta < \underline{\theta}^O$, but they are profitable when $\theta > \underline{\theta}^O$. Since $\tilde{e}^O(z) > \tilde{e}^M(z)$, as shown in Figure 1, the cost function of chemical fertilizer application is flatter, so $\underline{\theta}^O > \underline{\theta}^M$.

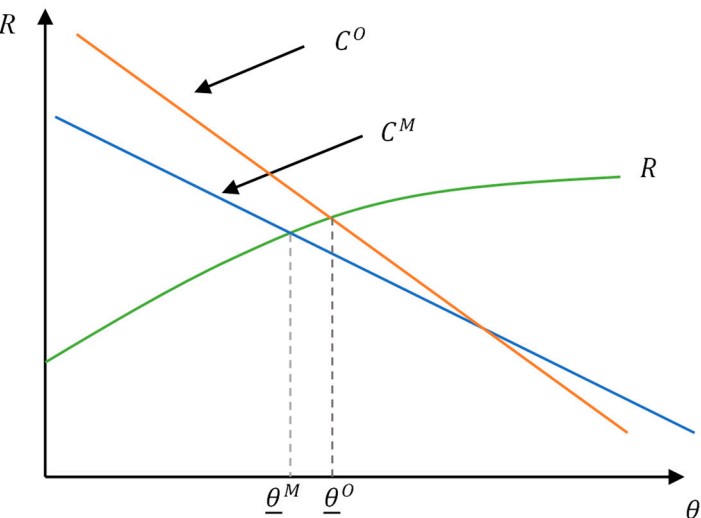

**Figure 1.** Cost function and income function of applying organic fertilizer and chemical fertilizer.

Figure 2 shows a case in which the revenue function changes. The higher risk of applying organic fertilizer leads to a reduction in the expected income. Hence, for farmers applying organic fertilizer, it is profitable when $\theta > \underline{\theta}^O$. Organic fertilizer application behavior requires higher $\theta$ than its chemical fertilizer equivalent, which implies $\underline{\theta}^O > \underline{\theta}^M$. Considering the changes in the cost and revenue functions described above, it can be proposed that due to the increased risk associated with the application of organic fertilizer, higher risk-prevention ability is required to achieve profitable agricultural production. The above-discussed anticipated rise in costs and reduction in revenue are expectations held by farmers that influence their decision-making process rather than being the actual sources of costs and revenue.

This section also attempts to analyze farmers' decision-making process in contract-farming participation. Theoretically, risk influences farmers' participation in contract farming [24]. Let us suppose that the revenue function remains constant and the cost function changes. Participants in contract farming have a cost function of $C^C(\cdot; \theta)$, and non-participants have a cost function of $C^N(\cdot; \theta)$. The costs of organic and chemical fertilizer application for farmers participating in contract farming are represented by $C^{OC}(\cdot; \theta)$ and

$C^{MC}(\cdot;\theta)$, respectively. Then, the cost of organic fertilizer application for contract-farming participants can be expressed as:

$$C^{OC}(\cdot;\theta) = (\alpha - \beta\theta^{OC})O \tag{5}$$

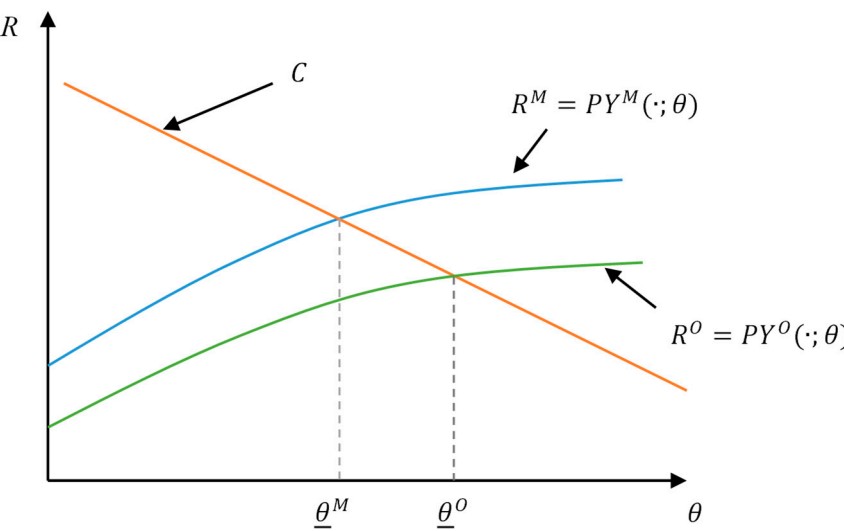

**Figure 2.** Cost function and revenue function of applying organic fertilizer and chemical fertilizer.

It is assumed that participants and non-participants in contract farming have similar qualitative properties of their cost functions, which means that $C_O, C_M$ are only related to $\theta$. Different ways to plot the cost function and the revenue function lead to different results, which depend on the location of $\underline{\theta}$ and $\underline{\theta}^C$. Contract farming may or may not be beneficial to farmers with all characteristics. Therefore, Figure 3 shows only an intermediate case and is consistent with our empirical analysis that participation or non-participation in contract farming is affected by specific subdivisions of farmer characteristics.

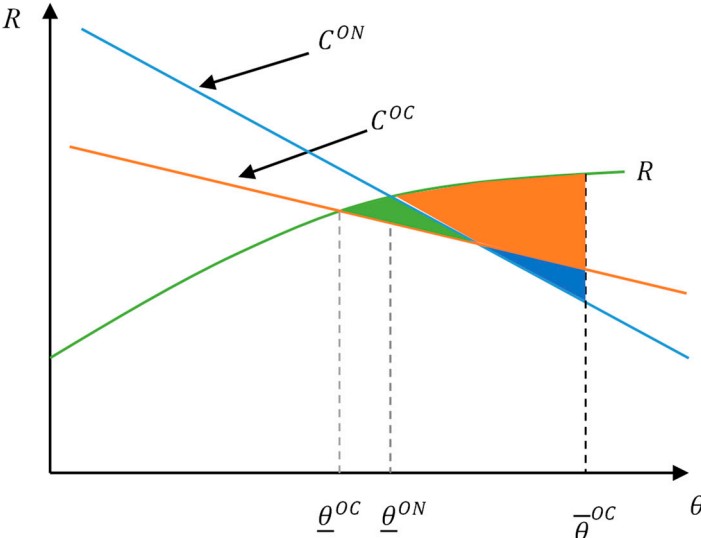

**Figure 3.** Cost and revenue functions of organic fertilizer use by contract-farming and non-contract-farming farmers.

As shown in Figure 3, when only organic fertilizer application is considered, farmers participating in contract farming must be at least $\underline{\theta}^{OC}$ to engage in farming. Non-contract-farming agricultural production brings better profits if its characteristics are higher than $\overline{\theta}^{OC}$. When their characteristics fall into $[\underline{\theta}^{OC}, \overline{\theta}^{OC}]$, farmers participate in contract farming,

because contract farming can bring higher profits. In Figure 4, the orange field represents the profits earned by all farmers. The green area represents the additional profits that farmers who participate in contract agriculture receive compared with those who do not participate in contract farming. The blue area represents the additional profits that farmers who do not participate in contract farming receive compared with those who participate in contract farming. Contract-farming participation reaches its upper limit when the blue area equals the green area.

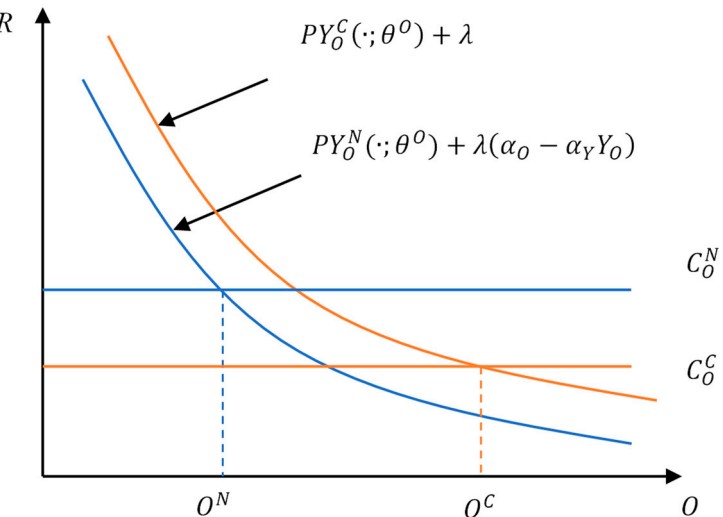

**Figure 4.** Solution to the Hamiltonian's optimality condition for organic fertilizer use.

Therefore, for farmers with lower risk-prevention ability, larger $\widetilde{e}(z)$ implies larger $\theta$. Given that the farming profitable range of $\theta$ ($[\underline{\theta}^{OC}, \overline{\theta}^{OC}]$) is wider when participating in contract farming than in non-contract farming ($[\underline{\theta}^{ON}, \overline{\theta}^{OC}]$), it is more likely that farmers with lower risk-prevention ability engage in contract farming. Hence, Hypothesis 1 can be proposed as shown below.

**Hypothesis 1.** *Farmers with lower risk-prevention ability are more willing to participate in contract farming.*

Next, we examine how contract farming influences a farmer's fertilizer application behavior. While chemical fertilizer is considered a static input, organic fertilizer can improve soil quality over time. Therefore, we hypothesize that the application of organic fertilizer improves soil quality with $\alpha_O$, while farming reduces soil quality by crop harvesting with $\alpha_Y$, with $\alpha_O > 0$ and $\alpha_Y > 0$. The evolution of soil quality over time can be expressed as:

$$\dot{S} = \alpha_O O - \alpha_Y Y(\cdot; \theta), S(0, \theta) = S_0 \tag{6}$$

where the dot over the variable represents the operator $d/dt$ and $S_0$ is the initial quality of the soil. Let us assume that the present value of soil quality at time $T$ is $S(T; \theta)e^{-\delta T}$, where $\delta$ represents the discount rate. We consider dynamic optimization within a finite period $T$, maximizing the present value of profits throughout the planning period.

$$J^* = \max_{O,M,\xi} \int_0^T [\xi PY^C + (1-\xi)PY^N - \xi C^C - (1-\xi)C^N]e^{-\delta t}dt + S(T;\theta)e^{-\delta T}$$

$$= \max_{O,M,\xi} \int_0^T \{\xi[PY^{OC}(\cdot;\theta^O) + PY^{MC}(\cdot;\theta^M)] + (1-\xi)[PY^{ON}(\cdot;\theta^O) + PY^{MN}(\cdot;\theta^M)] - \xi[(\alpha - \beta\theta^{OC})O \tag{7}$$

$$+ (\alpha - \beta\theta^{MC})M] - (1-\xi)[(\alpha - \beta\theta^{ON})O + (\alpha - \beta\theta^{MN})M]\}e^{-\delta t}dt + S(T;\theta)e^{-\delta T}$$

$$s.t. O > 0, M > 0, \theta > 0, \dot{S} = \alpha_O O - \alpha_Y Y(\cdot;\theta), S(0,\theta) = S_0, \tag{8}$$

where $O$, $M$, and $\xi$ are the control variables, and $S$ represents the state variables. We build the following Hamiltonian function:

$$H = \xi PY^C(\cdot;\theta) + (1-\xi)PY^N(\cdot;\theta) - \xi C^C(\cdot;\theta) - (1-\xi)C^N(\cdot;\theta) + \lambda(\alpha_O O - \alpha_Y Y(\cdot;\theta)) \quad (9)$$

We solve the Hamiltonian function as follows:

$$H_O = \xi PY_O^C(\cdot;\theta) + (1-\xi)PY_O^N(\cdot;\theta) - \xi C_O^C - (1-\xi)C_O^N + \lambda(\alpha_O - \alpha_Y Y_O) = 0 \quad (10)$$

$$H_M = \xi PY_M^C(\cdot;\theta) + (1-\xi)PY_M^N(\cdot;\theta) - \xi C_M^C - (1-\xi)C_M^N - \lambda \alpha_Y Y_M = 0 \quad (11)$$

$$H_\xi = PY^C(\cdot;\theta^O) - PY^N(\cdot;\theta^M) - C^C + C^N = 0 \quad (12)$$

$$\lambda' = -H_S = -\xi PY_S(\cdot;\theta^O) - (1-\xi)PY_S(\cdot;\theta^M) + \lambda \alpha_Y Y_S \quad (13)$$

$$S' = H_\lambda = \alpha_O O - \alpha_Y Y(O,M,S;\theta) \quad (14)$$

$$S(0,\theta) = S_0 \quad (15)$$

$$\lambda(T) = \frac{dS(T)e^{-\delta T}}{dS} \quad (16)$$

The Hamiltonian function is linear with respect to $\xi$ and $\xi \in [0,1]$. Therefore, the extremum occurs at the critical point of $\xi$ if the profit function is strictly increasing. The potential solutions for farmers are to choose either to participate or not to participate in contract farming.

Let us assume that the initial soil quality is the same among farmers; the steady state of soil quality is $S^\infty$, and the soil quality ($S(t)$) varies according to different decisions made by farmers regarding the application of organic and chemical fertilizers. Organic fertilizer application ($O$) is a substitute for chemical fertilizer application ($M$); $Y_{MS} < 0$, and $Y_{OS} < 0$. Therefore, as the soil quality ($S(t)$) increases, the marginal productivity of $O$ and $M$ decreases. Based on the above assumptions, farmers' optimal organic fertilizer application behaviors are given by Equation (10), as shown in Figure 4.

The blue line in Figure 4 provides the solution to Equation (10). Specifically, under the contract-farming participation condition, where $\xi = 1$, Equation (10) can be rewritten as:

$$H_O = PY_O^C(\cdot;\theta) - C_O^C + \lambda(\alpha_O - \alpha_Y Y_O) = 0 \quad (17)$$

Under the contract-farming non-participation condition, where $\xi = 0$, Equation (10) can be rewritten as:

$$H_O = PY_O^N(\cdot;\theta) - C_O^N + \lambda(\alpha_O - \alpha_Y Y_O) = 0 \quad (18)$$

Under the contract-farming participation condition, the derivative of the cost function with respect to $O$ is:

$$C_O^C = \alpha - \beta \theta^C = \alpha - \beta \theta^{OC} = \alpha - \beta \varphi \tilde{e}^{OC}(z) \quad (19)$$

Under the contract-farming non-participation condition, the derivative of the cost function with respect to $O$ is:

$$C_O^N = \alpha - \beta \theta^N = \alpha - \beta \theta^{ON} = \alpha - \beta \varphi \tilde{e}^{ON}(z) \quad (20)$$

Extensive empirical evidence suggests that contract farming can mitigate risks for farmers. Firstly, it directly reduces risks tied to the use of organic fertilizer in farming processes. Farmers engaged in contract farming often receive technical support from the contract issuer (e.g., IOFs or cooperatives). Those with robust risk-prevention ability and

knowledge of organic fertilizer use perceive lower or even no cost increase. Conversely, farmers with weaker risk-prevention skills and less familiarity with organic fertilizer use perceive a higher cost increase. Thus, contract-farming participation reduces the uncertainty of perceived cost increases. Secondly, contract farming alleviates inherent agricultural risks, including price volatility due to long production cycles. With a pre-agreed selling price, it reduces uncertainty due to price fluctuations and transaction costs due to price negotiations. It also mitigates market risks for small farmers in large markets by ensuring sales channel certainty, reducing the transaction costs of finding buyers, and lessening asset-specificity risks and the risks of fresh-produce spoilage. By agreeing on price and sales channels, it reduces market risks in specialized production [25,26]. Therefore, it is reasonable to believe that contract-farming participation can reduce the risks associated with organic fertilizer use, either by directly mitigating these risks or by reducing overall risk perception caused by inherent risks. This leads to $\tilde{e}^{OC}(z) < \tilde{e}^{ON}(z)$. Therefore, $C_O^C < C_O^N$ is illustrated in Figure 4 as the derivative of the cost function for contract farming, with respect to $O$, below the derivative of the cost function for non-contract farming, with respect to $O$. Likewise, farmers expect the organic fertilizer marginal output to be higher when they participate in contract farming due to the lower risks. Hence, $PY_O^C(\cdot; \theta) + \lambda(\alpha_O - \alpha_Y Y_O) > PY_O^N(\cdot; \theta) + \lambda(\alpha_O - \alpha_Y Y_O)$ is illustrated in Figure 4 as the orange line (participation in contract farming) above the blue line (non-participation in contract farming). $O^N$ and $O^C$ represent the optimal organic fertilizer use scenarios for farmers not participating and participating in contract farming, respectively. In light of this, research Hypothesis 2 is proposed below.

**Hypothesis 2.** *Participation in contract farming can increase farmers' organic fertilizer application behavior and decrease farmers' chemical fertilizer application behavior.*

## 3. Empirical Specification

Building on the proposed hypothesis and illustrated graph, we now move to a more detailed discussion on the empirical specification of the problem. In Equation (7), we assume that farmers with characteristics $\theta$ maximize the present value of expected farm profits over the planning period. However, the expected present value of farm profits is subjective and unobservable; what can be observed is the farmer's decision to participate in contract farming and the decision to apply organic fertilizer or chemical fertilizer. Therefore, we denote the unobservable latent variable, which is the expected present value of farm profits, by $R_{ik}^*$. We denote a farmer's decision to apply fertilizer by $R_{ik}$. When $R_{ik} = 1$, the farmer decides to use organic fertilizer or chemical fertilizer, and when $R_{ik} = 0$, the farmer decides not to use organic or chemical fertilizer. We refer to organic fertilizer when the subscript $k$ equals 1 and to fertilizer when $k$ equals 2. According to the maximization problem shown in Equation (7), $R_{ik}^*$ is positive when both $\partial J^*/\partial O$ and $\partial J^*/\partial M$ are positive. Furthermore, Equations (10) and (11) indicate that farmers' decisions to apply fertilizer are influenced by their decision to participate in contract farming, as well as by characteristics at the household and farm levels.

The equation for determining whether an individual farmer engages in contract farming may be expressed as an equation for the following latent variable:

$$
\begin{aligned}
\xi_i^* &= \beta Z_i + \varepsilon_i \\
\xi_i &= 1, \text{ if } \xi_i^* > 0, \\
\xi_i &= 0, \text{ if } \xi_i^* < 0,
\end{aligned}
\tag{21}
$$

where $\xi_i$ is a binary variable. If the expected present value of the profits from participating in contract farming, $\xi_{i1}^*$, is greater than the expected present value of the profits from non-contract farming, $\xi_{i0}^*$, which means that $\xi_i^* = \xi_{i1}^* - \xi_{i0}^* > 0$, then the farmer participates in contract farming, and $\xi_i = 1$. $Z_i$ represents the vector of factors influencing a farmer's decision to participate in contract farming. $\beta$ denotes the parameters to be estimated. $\varepsilon_i$ denotes the random error term assumed to be normally distributed.

The use of fertilizer by a farmer can be explained as follows:

$$
\begin{aligned}
R_{ik}^* &= \omega \xi_i + \gamma \theta_i + \mu_{ik}, \\
R_{ik} &= 1, \text{ if } R_{ik}^* > 0, \\
R_{ik} &= 0, \text{ if } R_{ik}^* \leq 0,
\end{aligned}
\tag{22}
$$

where $R_{ik}$ is a binary variable. When the farmer's expected profit present value, $R_{ik}^*$, is positive, farmer $i$ chooses to apply organic fertilizer ($k = 1$) or chemical fertilizer ($k = 2$), at which point $R_{ik}$ equals 1; otherwise, $R_{ik}$ equals 0. $\xi_i$ is a binary variable representing whether farmer $i$ participates in contract farming. $\omega$ and $\gamma$ are parameters to be estimated. $\mu_{ik}$ is the error term assumed to be normally distributed.

If the random error in Equations (21) and (22) contains the same unobservable variables, a selection bias occurs, resulting in a correlation between $\varepsilon_i$ and $\mu_{ik}$, which implies $corr(\varepsilon_i, \mu_i) = \rho_{\varepsilon\mu}$. To obtain unbiased estimates and strictly estimate the impact of contract farming on farmers' decisions to apply fertilizer, endogeneity must be addressed.

The recursive bivariate probit (RBP) model uses the full information maximum likelihood (FIML) method, simultaneously estimating the contract-farming participation equation and the fertilizer application decision equation, allowing for the estimation of the average treatment effect and marginal effect of endogenous binary variables on binary dependent variables when unobservable variables exist. The RBP model allows for the existence of the same variables in vector $\theta_i$ and vector $Z_i$, but this model requires at least one exogenous instrumental variable to identify the probability of farmers participating in contract farming, and this instrumental variable is unrelated to the farmers' fertilizer application decisions. In this study's RBP model, the logarithm of the "distance from the farmer's residence to the tea market" was used as an instrumental variable [27–29]. The farther the farmer's home is from the tea market, the higher the transaction costs. The increase in transaction costs leads farmers to internalize these costs, leading to transactions in the form of contracts. This is also the fundamental theory behind the emergence of contract farming [30–32]. Additionally, from various perspectives, there should not be a causal relationship between the distance from a farmer's home to the tea market and the farmer's fertilizer application decision.

We also estimate the average treatment effect on the treated (ATT) using the method proposed by Chiburis et al. (2012) [33]:

$$
ATT = \frac{1}{N_\xi} \sum_{i=1}^{N_\xi} \left\{ \Pr(Y_{ik} = 1 | \xi_i = 1) - \Pr(Y_{ik} = 0 | \xi_i = 1) \right\}
\tag{23}
$$

where $N_\xi$ denotes the sample from the treatment group, $\Pr(Y_{ik} = 1 | \xi_i = 1)$ represents the possibility of using organic fertilizer/chemical fertilizer of farmers participating in contract farming predicted by observed samples, and $\Pr(Y_{ik} = 0 | \xi_i = 1)$ represents the possibility of not using organic fertilizer/chemical fertilizer of farmers not participating in contract farming predicted by observed samples.

## 4. Data, Variables, and Descriptive Statistics

### 4.1. Data

The data for this study were gathered with field surveys conducted by our project team in 2017 in Quanzhou, Nanping, and Ningde in Fujian province, and in 2018 in Yichang, Hubei province. These surveys were designed to gather information about local tea farmers' production and sales activities during 2016.

Our sampling method was multi-stage and random. We began by selecting Fujian and Hubei provinces as the focus of our surveys. These provinces are situated in the Jiangnan tea region, one of China's four major tea-producing areas and a significant contributor to global tea production. In 2019, Fujian province, renowned as the "world's home of Oolong tea," produced 439,900 tons of tea, which represented 15.8% of China's total tea production. Hubei province contributed an additional 352,500 tons or 12.7% of the national total. Combined, these two provinces accounted for nearly 30% of China's tea production.

Within each province, we selected areas known for their tea production. In Fujian province, the city of Ningde is the top tea producer, followed by the cities of Quanzhou and Nanping. These areas produce a variety of teas, including green, black, red, and white teas. Fujian is famous for its Tianshan green tea, Jinmin red tea, Anxi Tieguanyin, Huangjingui, Wuyi Da Hong Pao, Rougui, and Shuixian. In Hubei province, well-known teas include Qings brick tea, Enshi Jade Dew, Yihong tea, Laojun eyebrow tea, Yingshan Cloud Mist, Dengcun green tea, Longfeng tea, Xiashou Bifeng, Baokang Songzhen, and Caihua Maojian.

In each village, we randomly selected 25–30 tea farmers for our survey, including both those who participated in contract farming and those who did not. Our questionnaire covered a wide range of topics, such as tea cultivation, picking, processing, sales, storage, social services, quality safety control, and risk-prevention ability. We collected a total of 956 questionnaires. After excluding those with missing values and logical contradictions, we had 694 valid responses, representing 69.67% and 72.59% of the total of the provinces of Fujian and Hubei, respectively.

### 4.2. Variables and Descriptive Statistics

Independent variables: The independent variables in this study were the decision to participate in contract farming and the degree of participation in contract farming. The former is a binary variable, with 1 indicating participation and 0 indicating non-participation. As per our data, 33.89% of farmers participated in contract farming. The degree of participation is represented by the continuous variable "percentage of tea sold under contracts in total sales", with an average of 4.52%.

Dependent variables: The dependent variables included two binary variables—the decision to apply organic fertilizer and the decision to apply chemical fertilizer—and two continuous variables—the intensity of the organic fertilizer application and that of the chemical fertilizer application. The binary variables were assigned a value of 1 if the farmer applied the respective fertilizer and 0 if not. Our data show that 64.13% of farmers applied organic fertilizer and 75% applied chemical fertilizer. The continuous variables are represented by the per-acre cost of using each type of fertilizer in tea planting, taken in logarithmic form.

Control variables: This paper utilizes the farmers' risk-prevention ability as the principal control variable, because it is believed that the farmers' risk-prevention ability affects the decision to participate in contract farming. This variable is measured by the question of whether farmers' tea-based income is affected by low temperature climates. If unaffected, we assign a value of 1 to this variable, suggesting robust risk-prevention abilities. If affected, the variable is designated a value of 0, signifying weaker risk-prevention abilities. The motivation behind gauging risk-prevention skills in this manner is based on the vulnerability of tea plants to cold weather. However, the surveyed regions, primarily in the lower hilly regions of central and northern Fujian and southern Hubei, are situated in the Jiangnan tea area. Here, the subtropical monsoon climate, late frosts, and northern cold currents pose potential hazards. Consequently, mitigating low-temperature risk becomes a critical factor for tea farmers when determining various production decisions. Notably, in the tea production process, managing low-temperature risks is a crucial skill for tea farmers. Descriptive statistics reveal that nearly 38% of tea farmers' incomes remains unaffected by low-temperature climates. Furthermore, this paper acknowledges the work of Rondhi et al. (2020) on the factors influencing contract farming participation [34], and Ma et al.'s (2018) research on farmers' fertilizer application [22]. Factors considered include the head of the household's age, years of education, gender, family size, urban resettlement preference, access to subsidies and loans, land fragmentation, land rights certification, tea cultivation area, and tea production experience. The definition and descriptive statistics for each of these variables are detailed in Table 1.

**Table 1.** Definitions of variables and descriptive statistics.

| Type | Variables | Definition | Std. Dev. | Mean | Min | Max | Obs. |
|---|---|---|---|---|---|---|---|
| Independent variables | *Confarm_par* | A value of 1 if farmer participated in contract farming and 0 otherwise. | 0.4736 | 0.3389 | 0 | 1 | 956 |
| | *Confarm_rate* | Tea sold through contracts as a percentage of all sales (%). | 0.1951 | 0.0452 | 0 | 1 | 921 |
| Dependent variables | *Orgafer_app* | A value of 1 if farmer used organic fertilizer and 0 otherwise. | 0.4799 | 0.6413 | 0 | 1 | 906 |
| | *Ln_orgainten* | Expenditure on organic fertilizer (CNY/mu). | 2.7535 | 3.3479 | 0 | 8.1120 | 943 |
| | *Chemfer_app* | A value of 1 if farmer used chemical fertilizer and 0 otherwise. | 0.4332 | 0.7500 | 0 | 1 | 940 |
| | *Ln_cheminten* | Expenditure on chemical fertilizer (CNY/mu). | 2.6289 | 4.2590 | 0 | 9.6159 | 943 |
| Control variables | *Riskpre* | A value of 1 if farmer's income from selling tea was affected by cold weather and 0 otherwise. | 0.4857 | 0.3796 | 0 | 1 | 548 |
| | *Ln_subsidy* | Subsidies given by the government for tea production and processing (CNY). | 3.0170 | 1.6334 | 0 | 12.2061 | 898 |
| | *Loan* | A value of 1 if farmer had loans and 0 otherwise. | 0.4100 | 0.2135 | 0 | 1 | 951 |
| | *Ln_landfrag* | Number of tea cultivation plots. | 0.6557 | 1.8328 | 0 | 6.3986 | 943 |
| | *Landcerti* | A value of 1 if farmer's land ownership had certification and 0 otherwise. | 0.3854 | 0.8188 | 0 | 1 | 949 |
| | *Area* | Area of tea cultivation land (mu). | 68.0715 | 19.9706 | 0 | 1000 | 951 |
| | *Farmexpe* | Tea industry experience of the longest-serving family member (years). | 15.6520 | 25.7692 | 0 | 32 | 953 |
| | *Age* | Age of the head of the household. | 11.2830 | 54.3868 | 21 | 90 | 954 |
| | *Edu* | Years of education of head of household. | 3.3578 | 7.4843 | 0 | 20 | 953 |
| | *Gender* | A value of 1 if the gender of head of household was male and 0 otherwise. | 0.2817 | 0.9132 | 0 | 1 | 956 |
| | *Familysize* | Number of members who stayed more than six months at home on a household's hukou. | 1.7198 | 4.3291 | 1 | 13 | 954 |
| | *Citywill* | A value of 1 if household was willing to give up agriculture and settle in the city and 0 otherwise. | 0.4164 | 0.2229 | 0 | 1 | 951 |
| Instrumental variables | *Marketdis* | Distance from farm to tea market (km). | 26.6603 | 4.6218 | 0 | 500 | 742 |
| | *Confarm_nei* | A value of 1 if there was a neighbor participating in contract farming and 0 otherwise. | 0.4926 | 0.4131 | 0 | 1 | 949 |

Instrumental variables: The instrumental variable used in this study was the "distance from the farmer's home to the tea market" in logarithmic form. The idea is that the farther the distance is, the higher the transaction cost of selling tea is, which increases the incentive to participate in contract farming. However, this distance should not affect the farmer's decision to apply fertilizer, making it an exogenous variable suitable as an instrumental variable. For robustness checks, the variable "whether the neighbor participates in contract farming" was used as an instrumental variable, based on the "peer effect".

## 5. Results and Discussion

In this section, we focus on the estimated results of the recursive bivariate probit (RBP) model. As a preliminary measure of the statistical validity of the RBP model, estimates of a seemingly unrelated bivariate probit (SUBP) model are presented first, followed by the goodness-of-fit test.

### 5.1. Results of SUBP Estimates

A SUBP model is mainly used to examine whether the decision to engage in contract farming relates to the outcome variables (fertilizer use) through unobserved heterogeneities, and whether these two decisions are substitutes or complementary. The estimation results of the SUBP model are shown in Table 2. Model 1 estimated the impact of the contract-farming participation decision on organic fertilizer application. Model 2 estimated the impact of the contract-farming participation decision on the chemical fertilizer application decision. The *p*-values of $\rho'_{\varepsilon\mu}$ in both Model 1 and Model 2 rejected the null hypothesis, thus confirming that $\rho'_{\varepsilon\mu} \neq 0$ was significant. The *p*-value indicated that the unobservable factors in the decision to participate in contract farming were correlated with the unobservable factors in the decision to apply organic or chemical fertilizers, which confirmed the presence of

endogeneity. The impact of the unobservable factors captured by the error term suggests that the likelihood of a farmer's choice to participate in contract farming was related to their choice to apply organic or chemical fertilizer. Moreover, $\rho'_{\varepsilon\mu}$ was positive in Model 1, implying that participating in contract farming and applying organic fertilizer are complementary decisions, while $\rho'_{\varepsilon\mu}$ was negative in Model 2, suggesting that participating in contract farming and applying chemical fertilizer are substitute decisions [35,36].

**Table 2.** Results of SUBP model estimates.

| Variables | Model 1 | | Model 2 | |
|---|---|---|---|---|
| | Participation | Organic Fertilizer | Participation | Chemical Fertilizer |
| Riskpre | −0.3767 ** | | −0.4205 ** | |
| | (0.1331) | | (0.1326) | |
| Age | −0.0722 * | 0.0162 | −0.0696 * | −0.0333 |
| | (0.0333) | (0.0344) | (0.0320) | (0.0340) |
| Age_square | 0.0006 * | −0.0003 | 0.0006 * | 0.0002 |
| | (0.0003) | (0.0003) | (0.0003) | (0.0003) |
| Edu | −0.0085 | 0.1135 * | −0.0354 | −0.0392 |
| | (0.0499) | (0.0466) | (0.0451) | (0.0481) |
| Edu_square | 0.0030 | −0.0079 * | 0.0043 | 0.0002 |
| | (0.0033) | (0.0031) | (0.0030) | (0.0032) |
| Gender | 0.1645 | 0.3403 * | 0.2351 | −0.1530 |
| | (0.1853) | (0.1695) | (0.1809) | (0.1888) |
| Familysize | 0.0680 * | 0.0278 | 0.0475 | −0.0294 |
| | (0.0323) | (0.0334) | (0.0315) | (0.0325) |
| Ln_subsidy | 0.0119 | −0.0042 | 0.0062 | 0.0306 |
| | (0.0181) | (0.0175) | (0.0177) | (0.0190) |
| Loan | 0.1100 | 0.2918 * | 0.1296 | −0.0348 |
| | (0.1324) | (0.1322) | (0.1259) | (0.1298) |
| Ln_landfrag | −0.1791 * | 0.1871 * | −0.1749 | 0.1148 |
| | (0.0915) | (0.0909) | (0.0887) | (0.0913) |
| Landcerti | −0.0755 | −0.2506 | −0.0608 | 0.2268 |
| | (0.1320) | (0.1374) | (0.1299) | (0.1327) |
| Area | 14.7464 | 0.9959 | 15.2960 | −8.7925 |
| | (8.7753) | (7.7655) | (9.3908) | (7.5614) |
| Farmexp | −0.0095 * | 0.0027 | −0.0113 * | 0.0022 |
| | (0.0044) | (0.0032) | (0.0044) | (0.0031) |
| Citywill | −0.2686 | −0.3645 ** | −0.1815 | 0.2143 |
| | (0.1383) | (0.1243) | (0.1301) | (0.1344) |
| Marketdis | 0.0056 *** | | 0.0087 *** | |
| | (0.0007) | | (0.0009) | |
| Confarm_nei | 1.6367 | −0.6086 | 1.7061 | 1.7993 |
| | (0.9347) | (0.9704) | (0.9099) | (0.969) |
| $\rho'_{\varepsilon\mu}$ | 0.3627 *** | | −0.2037 ** | |
| Log-likelihood | −779.4842 | | −774.6875 | |
| Wald test:$\rho'_{\varepsilon\mu}$ | 24.6557 *** | | −0.2037 ** | |
| Sample size | 694 | | 694 | |

*, **, and *** indicate significance at the 10%, 5%, and 1% levels, respectively, with the robust standard error being shown in parentheses.

Additionally, to validate the suitability of the recursive binary probit (RBP) model, certain tests are needed [35]. This paper used the score test proposed by Murphy (2007) (Murphy score test) and the Hosmer–Lemeshow goodness-of-fit test suggested by Hosmer and Lemeshow (1980) to check whether the RBP model could perform a good maximization fit of the joint density of the dependent variables [37,38]. The results are shown in Table 3. The null hypothesis of the Murphy score test is that the error terms of the regression in the first and second stages are joint binary standard normal distributions. The null hypothesis of the Hosmer–Lemeshow goodness-of-fit test is that the sample frequency of the dependent

variable is the same as the goodness-of-fit probability of the observed subsample. The results show that all p-values were not significantly different from zero at the 10% level, and the null hypothesis was not rejected, meaning that the RBP model was valid.

**Table 3.** Murphy score test and Hosmer–Lemeshow suitability test.

| | **Murphy's Score Test** | **Hosmer–Lemeshow Test** |
|---|---|---|
| Contract farming and organic fertilizer application | Chi2(9) = 8.5200 with Prob > chi2 = 0.4830 | Chi2(9) = 18.6900 with Prob > chi2 = 0.6048 |
| Contract farming and chemical fertilizer application | Chi2(9) = 7.3100 with Prob > chi2 = 0.6235 | Chi2(9) = 22.6700 with Prob > chi2 = 0.3618 |

*5.2. Results of RBP Estimates*

Based on the RBP model, Table 2 shows the estimates of the determinants of contract-farming participation and their impact on organic fertilizer and chemical fertilizer application. In the RBP model, the FIML method was used to jointly estimate the con-tract-farming participation equation and the organic or chemical fertilizer application decision equation. The results presented in the lower part of Table 4 show that all estimated correlation coefficients, $\rho'_{\varepsilon\mu}$, in Models 1 and 2 no longer differ significantly from 0, suggesting that unobservable factors no longer contribute to endogeneity. In addition, the Wald tests of Models 1 and 2 did not reject the null hypothesis of being zero, indicating that participation in contract farming was an exogenous variable. The two decisions are not made simultaneously. Farmers first decide whether to participate in contract farming and then decide whether to apply organic or chemical fertilizer [39].

**Table 4.** Results of RBP model estimates.

| Variables | Model 1 | | Model 2 | |
|---|---|---|---|---|
| | **Participation** | **Organic Fertilizer** | **Participation** | **Chemical Fertilizer** |
| *Confarm_par* | | 1.8529 *** (0.1234) | | −1.5326 *** (0.3997) |
| *Riskpre* | −0.3356 * (0.1344) | | −0.3390 * (0.1424) | |
| *Controls* | Yes | Yes | Yes | Yes |
| *Instruments* | Yes | Yes | Yes | Yes |
| *Constant* | 0.7593 (1.4033) | −4.0068 *** (1.1363) | 1.2856 (1.5757) | 3.4315 * (1.3731) |
| $\rho'_{\varepsilon\mu}$ | | −5.2820 (8.1792) | | 0.7671 (0.4067) |
| *Log-likelihood* | | −766.83 | | 765.65 |
| *Wald test:*$\rho'_{\varepsilon\mu}$ | | 0.4170 | | 3.5584 |
| *ATE* | | 0.4820 *** (0.1424) | | −0.5242 *** (0. 1341) |
| *ATT* | | 0.7927 *** (0.2176) | | −0.3695 *** (0. 1311) |
| *Sample size* | | 666 | | 694 |

* and *** indicate significance at the 10% and 1% levels, respectively, with the robust standard error being shown in parentheses.

*5.3. Determinants of Contract-Farming Participation and Fertilizer Application*

As a result of the RBP model, Table 4 displays the impact of contract farming participation on farmers' fertilizer application. This consists of two stages.

Firstly, in Table 4, the second and fourth columns present the results of the first-stage estimations of the RBP model, which identify the determinants of farmers' decisions to engage in contract farming. The estimated results of Model 1 and Model 2 are similar in terms of their coefficient signs and significance levels, so the two models are discussed

together. Farmers' risk-prevention abilities significantly deterred contract-farming participation, suggesting that farmers with weaker risk-prevention abilities were more inclined to participate in contract farming, while those with stronger risk-prevention ability were less likely to engage. This result validates Hypothesis 1 of this paper, supporting the idea that risk management at the farm level is a significant factor for farmers [40,41]. This finding is also consistent with the widely accepted view in the literature that contract farming serves a risk-shifting function [25,26,41,42]. From an empirical standpoint, this paper presents evidence suggesting that farmers' risk-prevention abilities significantly influences farmers' participation in contract farming. It is necessary to point out that in the literature concerning risk as a determinant of contract-farming participation, risk is mainly measured using risk preferences [43–45]. This result provides an alternative perspective, because risk-prevention ability is the outcome variable, while risk preference is the latent variable. The results show that risk factors affecting contract farming can be measured using farmers' risk-prevention ability. This solves the problem that risk preference is difficult to objectively quantify to a certain extent.

Secondly, the third and fifth columns of Table 4 present the results of the second stage of the RBP model, that is, the factors influencing farmers' organic and chemical fertilizer application behavior, respectively. Based on the estimated results in the third column, contract farming had a significant positive impact on farmers' organic fertilizer application behavior. The fifth column shows the impact of participation in contract farming on farmers' chemical fertilizer application behavior. The coefficient of contract-farming participation was significantly negative, indicating that participation in contract farming curbed the decision to apply chemical fertilizers. Based on this result, Hypothesis 2 in this paper is confirmed. It is consistent with the existing literature that contract farming is beneficial for adopting production techniques [46–48]. It confirms the view that contract farming contributes to an increase in the application of organic fertilizer and a reduction in fertilizer use [20,48]. However, the existing literature does not fully consider the heterogeneity of farmers' participation in contract agriculture [48], nor does it consider the role of risk in farmers' participation in contract agriculture [20]. As a result of the study, empirical evidence is provided supporting the role of contract agriculture in farmers adopting green production technologies [49,50], indicating that contract agriculture contributes to the sustainable development of agriculture and the achievement of the global sustainable development goals (SDGs).

The results demonstrate that a farmer's risk-prevention ability influences their decision to participate in contract farming, which, in turn, affects their choice to use organic or chemical fertilizer. By incorporating the risk-prevention ability into the decision-making model, we can better understand how contract farming influences farmers' fertilizer application behavior. Specifically, farmers with lower risk-prevention ability are more likely to engage in contract farming as a risk-mitigation strategy, which can enhance their production behaviors, such as increasing the use of organic fertilizer and reducing the use of chemical fertilizer. Furthermore, the promotion of organic fertilizer use and the reduction in chemical fertilizer use in contract farming suggest a substitution relationship between the decisions to apply these two types of fertilizer.

### 5.4. Marginal Effects and Average Treatment Effects

Considering that the estimated coefficients of the explanatory variables in Table 4 cannot be directly interpreted, we also calculated the marginal effects to better under-stand the impact of the variables on investment. The results in Table 5 show that participation in contract farming increased the probability of farmers using organic fertilizer by 48.13% but reduced the probability of chemical fertilizer use by 53.24%. Among other variables, for each additional plot of land, the likelihood of using organic fertilizer decreased by 8.72%, and the likelihood of using chemical fertilizer decreased by 4.48%. For every additional acre of tea cultivation area, the likelihood of using organic fertilizer increased by 0.64%, and the likelihood of using chemical fertilizer increased by 0.37%.

**Table 5.** Marginal effects of RBP model estimation on marginal probability.

| Variables | Organic Fertilizer | Chemical Fertilizer |
|---|---|---|
| *Confarm_par* | 0.4813 | −0.5324 |
| *Age* | −0.0016 | −0.0204 |
| *Age_square* | 0.0000 | 0.0002 |
| *Edu* | −0.0282 | −0.0032 |
| *Edu_square* | 0.0012 | 0.0002 |
| *Gender* | 0.1276 | 0.0654 |
| *Familysize* | −0.0129 | −0.0103 |
| *Ln_subsidy* | 0.0111 | 0.0076 |
| *Loan* | 0.0220 | 0.0019 |
| *Ln_landfrag* | −0.0872 | −0.0448 |
| *Landcerti* | 0.0881 | 0.0927 |
| *Area* | 0.0064 | 0.0037 |
| *Farmexpe* | 0.0010 | 0.0001 |
| *Citywill* | −0.0596 | −0.0138 |

We also adopted the method proposed by Chiburis et al. (2012) to estimate the average treatment effect (ATE) and the average treatment effect on the treated (ATT), as shown in Table 4 [33]. This method uses bootstrap replications (sampling with replacement from the original sample) to reduce the sample noise. The results show that for all samples, participating in contract farming increased the likelihood of using organic fertilizer by 48.20%. For the treated group, compared with not participating in contract farming, participating in contract farming significantly increased the probability of farmers using organic fertilizer, with an increase of 79.27%. For the full sample, participating in contract farming reduced the likelihood of farmers using chemical fertilizers by 52.42%. For the treated group, compared with not participating in contract farming, participation also reduced the likelihood of farmers using chemical fertilizers by 36.95%. Although there was a slight difference in numerical values between the marginal effects estimated by the RBP model and the average treatment effects, they both showed a strong and significant impact of participating in contract farming on the decision to use organic and chemical fertilizers.

*5.5. Robustness Test*

For robustness checks, we took two measures. First, we applied the two-stage least squares (2SLS) method. This method helped us transform binary variables into continuous ones, allowing us to view the decision to participate in contract farming, not just as a yes or no choice but rather as a matter of degree or extent of participation. Similarly, the decision to apply organic and chemical fertilizers was expanded to consider the intensity of fertilizer application. Specifically, we converted "whether to participate in contract farming" into "the percentage of tea sold through contract farming," and "whether to use organic/chemical fertilizers" into "average organic/chemical fertilizer expenditure per acre". Secondly, we replaced the instrumental variables, switching from the original distance from the tea sales market to "whether the neighbors participate in contract farming". This variable should not affect the farmers' behavior of applying organic and chemical fertilizers, but according to the "peer effect," it may influence the likelihood of farmers' participating in contract farming.

As shown in Table 6, both the second and fourth columns represent the impact of individual farmer factors, such as risk-prevention ability abd contract-farming decisions, while the third and fifth columns represent the impact of contract-farming decisions on organic and chemical fertilizer application intensity, respectively. The instrumental variable in both Model 1 and Model 2 was whether the neighbors participated in contract farming. Both models passed the unidentifiability test, the weak identification test, and the Hansen J exogeneity test. In the first stage of the model, namely, columns 2 and 4, the results are the same. The results show that farmers with strong risk-prevention ability were less likely to participate in contract farming, which is consistent with the results obtained with the

previous SUBP model. The results in column 3 show that participation in contract farming could increase organic fertilizer application intensity. Specifically, farmers who participated in contract farming spent CNY 4.26 more per acre on organic fertilizer than those who did not participate in contract farming. The results in column 5 indicate that participating in contract farming could reduce chemical fertilizer application. Specifically, farmers who participated in contract farming spent CNY 2.44 less per acre on chemical fertilizers than those who did not participate in contract farming.

**Table 6.** The 2SLS model estimation: contract-farming participation and fertilizer application intensity.

| Variables | Model 1 | | Model 2 | |
|---|---|---|---|---|
| | Participation | Organic Fertilizer Application Intensity | Participation | Chemical Fertilizer Application Intensity |
| Confarm_par | | 4.2624 *** (1.1968) | | −2.4434 * (1.0984) |
| Riskpre | −0.0797 * (0.0367) | | −0.0797 * (0.0367) | |
| Controls | Yes | Yes | Yes | Yes |
| Instruments | Yes | Yes | Yes | Yes |
| Obs. | | 502 | | 502 |
| Underidentification test | | 32.8720 *** | | 32.8720 *** |
| Kleibergen–Paap rk LM statistic | | [0.0000] | | [0.0000] |
| Weakidentification test | | 18.0430 | | 18.0430 |
| Kleibergen–Paap rk Wald F statistic | | {11.59} | | {11.59} |
| Overidentification test | | 0.0060 | | 2.7460 |
| Hansen J statistic | | [0.9403] | | [0.0975] |

* and *** indicate significance at the 10% and 1% levels, respectively, with the robust standard error being shown in parentheses. The *p*-values are in square brackets. The critical values corresponding to the Stock–Yogo test at the 15% level are in curly brackets.

Table 7 shows the robustness test after converting the decision to use organic fertilizer and chemical fertilizer into the degree of organic and chemical fertilizer use and after converting the decision to participate in contract farming into the degree of participation in contract farming. Both models passed the underidentification test, the weak identification test, and the Hansen J exogeneity test. The results show that the degree of participation in contract farming, namely, the percentage of tea sold through contract farming, was not affected by the risk-prevention ability and did not affect the degree of organic and chemical fertilizer use.

Table 8 shows the impact of the contract participation degree on the decision to use organic and chemical fertilizers. Both models passed the underidentification test, the weak identification test, and the Hansen J exogeneity test. The results indicate that the degree of participation in contract farming was not affected by the ability to prevent risks and that the degree of participation in contract farming did not affect the decision to use organic and chemical fertilizers. Combining the results of Tables 7 and 8, we believe that the degree of participation in contract farming is not a key factor affecting farmers' decisions and the degree of organic fertilizer use. Instead, the key factor lies in the decision to participate in contract farming—what is commonly known as "participation is key". The possible reason is that as long as farmers participate in contract farming, the psychological level of risk shift can be achieved. Therefore, the level of participation no longer matters.

**Table 7.** The 2SLS model estimation: contract-farming participation rate and fertilizer application intensity.

| Variables | Model 1 | | Model 2 | |
|---|---|---|---|---|
| | Participation Rate | Organic Fertilizer Application Intensity | Participation Rate | Chemical Fertilizer Application Intensity |
| *Confarm_rate* | | 1.5810 | | −2.3703 |
| | | (3.2889) | | (1.5419) |
| *Riskpre* | −0.0044 | | −0.0044 | |
| | (0.0185) | | (0.0185) | |
| *Controls* | Yes | Yes | Yes | Yes |
| *Instrumentals* | Yes | Yes | Yes | Yes |
| *Obs.* | 388 | | 388 | |
| *Underidentification test* | 9.8010 * | | 9.8010 * | |
| *Kleibergen–Paap rk LM statistic* | [0.0203] | | [0.0203] | |
| *Weak-variable-identification test* | 23.5130 | | 23.5130 | |
| *Kleibergen–Paap rk Wald F statistic* | {13.9100} | | {13.9100} | |
| *Overidentification test* | 5.8790 | | 7.3390 | |
| *Hansen J statistic* | [0.05290] | | [0.0255] | |

* indicate significance at the 10% levels, with the robust standard error being shown in parentheses. The *p*-values are in square brackets. The critical values corresponding to the Stock–Yogo test at the 15% level are in curly brackets.

**Table 8.** The 2SLS model estimation: contract-farming participation rate and fertilizer application.

| Variables | Model 1 | | Model 2 | |
|---|---|---|---|---|
| | Participation Rate | Organic Fertilizer Application | Participation Rate | Chemical Fertilizer Application |
| *Confarm_rate* | | 0.0614 | | −0.3188 |
| | | (0.4910) | | (0.3130) |
| *Riskpre* | −0.0057 | | −0.0049 | |
| | (0.0188) | | (0.0187) | |
| *Controls* | Yes | Yes | Yes | Yes |
| *Instruments* | Yes | Yes | Yes | Yes |
| *Obs.* | 380 | | 380 | |
| *Underidentification test* | 10.138 * | | 10.215 * | |
| *Kleibergen–Paap rk LM statistic* | [0.0174] | | [0.0168] | |
| *Weak-variable-identification test* | 23.5750 | | 23.6300 | |
| *Kleibergen–Paap rk Wald F statistic* | {13.9100} | | {13.9100} | |
| *Overidentification test* | 0.0628 | | 5.7660 | |
| *Hansen J statistic* | [0.0528] | | [0.0560] | |

* indicate significance at the 10% levels, with the robust standard error being shown in parentheses. The p-values are in square brackets. The critical values corresponding to the Stock–Yogo test at the 15% level are in curly brackets.

### 5.6. Empirical Results Discussion

The study examines the relevance between contract farming participation and fertilizer application, showing that farmers with lower risk-prevention abilities are more likely to participate in contract farming. This participation increases the use of organic fertilizers and decreases the use of chemical ones. The decision to participate in contract farming is more influential on fertilizer use than the extent of participation. This paper contributes to filling that gap by linking the decision to participate in contract farming to shifts in fertilizer use. In line with previous findings, it underscores contract farming's function as a risk management strategy [25,26,41,42]. However, it further unveils a connection between farmers' risk-prevention abilities and their inclination to participate in contract farming, an aspect less considered in the literature [43,45]. Moreover, this study adds new insights by demonstrating that the decision to participate in contract farming has a significant impact on fertilizer use, contributing to the discourse on sustainable agricultural practices. This reflects and expands existing studies, which highlighted the potential of contract farming

in promoting sustainability [12–14]. The findings of this study offer a richer understanding of the complex dynamics at play in contract farming decisions and their environmental implications, thereby providing valuable empirical evidence to shape future research and policy in sustainable agriculture.

## 6. Conclusions

This study investigated the influence of contract-farming participation on farmers' utilization of organic fertilizer and reduction in chemical fertilizer usage. A dynamic model was developed, emphasizing farmers' risk-prevention ability and determining how contract-farming participation, and household-level and production-level characteristics impact fertilizer usage behaviors. We employed field survey data on tea farmers from Fujian and Hubei provinces, China, and analyzed their decision to use organic/chemical fertilizers and the degrees of organic/chemical fertilizer usage from a risk perspective. The study concluded that under limited household-level and production-level characteristics, it is optimal for farmers to participate in contract farming. Risk-prevention ability emerged as a key factor driving this decision. The findings also reveal that contract-farming participants were more likely to apply organic fertilizers and less likely to use chemical fertilizers.

Our research results advocate for policy measures encouraging small farmers to participate in contract farming. This policy has potential benefits, including the promotion of organic soil enhancement behaviors, reduction in chemical fertilizer usage, green agricultural development, sustainable soil fertility, and environmental protection. Furthermore, participation in contract farming could help mitigate the risks that farmers face when adopting new agricultural technologies, reduce transaction cost risks for small farmers in expansive markets, and alleviate the sales risks due to price fluctuations. Encouraging contract farming could play a significant role in promoting organic fertilizer use and minimizing chemical fertilizer usage among farmers.

While our study provides comprehensive insights, it also acknowledges certain limitations. The data only covered short-term cross-sectional aspects, limiting the verification of our theoretical analysis to short-term impacts. Our analysis did not consider the impact of the degree of contract-farming participation on fertilizer usage. Additionally, the recursive binary probit model results indicate that various factors, such as tea cultivation area, distance between residence and tea market, risk-prevention ability, plot number, land rights confirmation, and age, influenced decisions related to participation in contract farming and fertilizer usage. Future research should aim to gather long-term data and account for variations in contract-farming participation, offering a more extensive understanding of these relationships.

**Author Contributions:** Conceptualization, Y.Z. and R.Y.; methodology, Y.Z.; software, R.Y.; validation, K.Z., X.K. and Y.Z.; formal analysis, Y.Z.; investigation, Y.Z. and X.K.; resources, X.K.; data curation, Y.Z.; writing—original draft preparation, Y.Z.; writing—review and editing, K.Z. and X.K.; visualization, Y.Z.; supervision, R.Y., X.K. and K.Z.; project administration, K.Z.; funding acquisition, X.K. All authors have read and agreed to the published version of the manuscript.

**Funding:** This research received no external funding.

**Informed Consent Statement:** Informed consent was obtained from all subjects involved in the study.

**Data Availability Statement:** The datasets used and/or analyzed during the current study are available from the corresponding author upon reasonable request.

**Conflicts of Interest:** The authors declare no conflict of interest.

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
