# Peer review of "Fertilizer Application in Contract Farming: A Risk Analysis"

_land, doi:10.3390/land12081495_

Round 1
Reviewer 1 Report
Overall, the manuscript is very poorly written with a major requirement to rewrite several sections and restructure the whole manuscript. The equations used for the study are appropriate, however, they are poorly explained with their relevance difficult to follow. There is data stated in nearly every section, even though there is a results section. This is poor practice. It is unclear if the data reported supports the conclusion. The manuscript cannot continue any further until the major errors in structure are addressed.
Author affiliations need to be corrected. This should not just be email addresses. All authors need an affiliation.
Line 7: Is this is the first of it's study in the world or in China?
Line 13: Never start a sentence with 'and'
Line 26: How many is a 'few'? Give the number of decades.
Line 28: Where's 'here'? What country are you talking about?
Line 33: Needs a reference.
Line 34: Don't use 'etc'. This is lazy. Write your list out in full.
Line 34: Define 'CF'
Line 39: Needs a reference.
Lines 40 to 45: This should be at the end of the introduction.
The introduction needs many more references and improved structure.
Section 2 needs to be reworded. The tone of the section is too 'chatty' and not professional enough for a published article. The section also reads as a list of instructions from your notes rather than in the correct tense for a methods section.
Lines 120 and 121: Sentence structure needs to be addressed.
Line 132: Reference format.
20 references for this paper is not enough. There should be almost twice this many references.
Sentence structure and general grammar need to be addressed.
Reviewer 2 Report
Dear Sir
following thing need to be improved
##Gist is missing moreover, using data of tea farmers in Fujian and Hubei provinces in China, the recursive binary probit model was used to estimate the impact of contract farming participation on farmers' organic fertilizer and chemical fertilizer application decisions. And the 2SLS model was used to estimate the impact of contract farming participation rate on farmers' organic and chemical fertilizer application intensity. The empirical results show that the risk-prevention ability has a significant negative impact on farmers' contract farming participation decisions and rates.
## reframe the sentence In recent years, the potential of contract farming to foster sustainable agricultural 45 practices has been increasingly recognized [5]. The worldwide emphasis on sustainable 46 agriculture is driven by the growing understanding of the environmental, health, and economic implications of traditional farming methods, notably the excessive use of chemical fertilizers [6]. These chemical fertilizers, while boosting crop yield in the short term, have been linked to a variety of environmental issues such as soil degradation, water pollution, and loss of biodiversity. Moreover, the overreliance on chemical fertilizers often results indiminishing returns over time, with increased quantities needed to maintain productivity [7].
##. Check the modal equation At period t , Ot() and Mt() de- 130 note the possibility of a farmer’s choice of organic fertilizer and chemical fertilizer usage 131 respectively. The same as Ma et al.(2018), assuming farmers won’t change their decision 132 to contract farming participation. Let Y() denote the production function of one unit of 133 land cultivation, which is a function of chemical fertilizer usage Mt() , organic fertilizer 134 usage Ot() , soil quality St() and farmer’s characteristics . And Y() is strictly con- 135 cave in arguments O M S , , and additive separable
## colourful
Figure 1. Cost function and income function of applying organic fertilizer and chemical fertilizer.
Figure 2. Cost function and revenue function of applying organic fertilizer and chemical fertilizer.
## concise the Conclusion
Round 2
Reviewer 1 Report
This is much better. It is clear that a lot of work has gone into this.
There are a few instances in the introduction were the references are in the wrong format.
The final paragraph in the introduction 'This paper is organized ...' is not really needed.
Statistical results could be made a little clearer.
